# A Panel-Agnostic Strategy ‘HiPPo’ Improves Diagnostic Efficiency in the UK Genomic Medicine Service

**DOI:** 10.3390/healthcare11243179

**Published:** 2023-12-15

**Authors:** Eleanor G. Seaby, N. Simon Thomas, David Hunt, Diana Baralle, Heidi L. Rehm, Anne O’Donnell-Luria, Sarah Ennis

**Affiliations:** 1Human Development and Health, Faculty of Medicine, University Hospital Southampton, Southampton SO16 6YD, Hampshire, UK; david.hunt2@uhs.nhs.uk (D.H.); d.baralle@soton.ac.uk (D.B.); se@soton.ac.uk (S.E.); 2Program in Medical and Population Genetics, Broad Institute of MIT and Harvard, Cambridge, MA 02142, USA; hrehm@mgh.harvard.edu (H.L.R.); odonnell@broadinstitute.org (A.O.-L.); 3Division of Genetics and Genomics, Boston Children’s Hospital, Boston, MA 02115, USA; 4Paediatric Infectious Diseases, Imperial College London, London W2 1NY, UK; 5Wessex Regional Genomics Laboratory, Salisbury NHS Foundation Trust, Salisbury SP2 8BJ, UK; simon.thomas1@nhs.net; 6Center for Genomic Medicine, Massachusetts General Hospital, Boston, MA 02114, USA

**Keywords:** rare disease, genetics, rare disease analysis, genomics, gene-agnostic

## Abstract

Genome sequencing is available as a clinical test in the UK through the Genomic Medicine Service (GMS). The GMS analytical strategy predominantly filters genome data on preselected gene panels. Whilst this reduces variants requiring assessment by reporting laboratories, pathogenic variants outside applied panels may be missed, and variants in genes without established disease–gene relationships are largely ignored. This study compares the analysis of a research exome to a GMS clinical genome for the same patients. For the research exome, we applied a panel-agnostic approach filtering for variants with **Hi**gh **P**athogenic **Po**tential (HiPPo) using ClinVar, allele frequency, and in silico prediction tools. We then restricted HiPPo variants to Gene Curation Coalition (GenCC) disease genes. These results were compared with the GMS genome panel-based approach. Twenty-four participants from eight families underwent parallel research exome and GMS genome sequencing. Exome HiPPo analysis identified a similar number of variants as the GMS panel-based approach. GMS genome analysis returned two pathogenic variants and one de novo variant. Exome HiPPo analysis returned the same variants plus an additional pathogenic variant and three further de novo variants in novel genes, where case series are underway. When HiPPo was restricted to GenCC disease genes, statistically fewer variants required assessment to identify more pathogenic variants than reported by the GMS, giving a diagnostic rate per variant assessed of 20% for HiPPo versus 3% for the GMS. With UK plans to sequence 5 million genomes, strategies are needed to optimise genome analysis beyond gene panels whilst minimising the burden of variants requiring clinical assessment.

## 1. Introduction

Genome sequencing is now available as a diagnostic test on the National Health Service (NHS) in the UK, offered through their Genomic Medicine Service (GMS). With the cost of genome sequencing becoming ever-competitive, genome sequencing is beginning to supersede exome sequencing in some institutes, including in the NHS [1]. However, one of the challenges in diagnosing patients with rare diseases is the expanded scope of analysis and the need to correlate results with phenotype [2]. Genome sequencing produces 3–4 million variants per individual; therefore, strategies to reduce noise and focus on the most salient regions of DNA have been adopted, including the use of virtual gene panels [3,4]. For the NHS, this is their primary analytical strategy, meaning that despite sequencing and storing an entire genome, only a fraction of the genome is actually analysed. Consequently, this risks missing pathogenic variants that would have been identified if more regions of the genome had been assessed.

All that said, there remains a trade-off between utilising the breadth of sequencing data available (such as for a genome) and the number of variants that require assessment by clinical laboratories. Filtering is necessary to reduce the number of variants identified to a manageable number that NHS laboratories can analyse, classify with respect to pathogenicity, and interpret with respect to the causality of the patient’s symptoms in a reasonable and acceptable timeframe.

The GMS, which primarily sequences trios, adopts a workflow similar to that used in the 100,000 Genomes Project, which predated the GMS [1,5]. First, the data are filtered by inheritance pattern(s), data quality, and allele frequency. Following this, the remaining variants are filtered by a gene panel(s) selected by the clinician when the test is ordered, meaning that only coding regions are considered. Short variants and copy number variants (CNVs) overlapping the gene panel are returned for analysis (“Tiered variants”). The only variants mandated to be assessed outside of the gene panel(s) are ‘gene-agnostic variants’ comprising de novo coding variants and Exomiser [6] top three ranked variants, which are not filtered on quality (Figure 1).

In contrast to genome sequencing, exome sequencing targets only coding regions of DNA. However, most variants filtered in the GMS strategy (Tier 1, Tier 2, and gene-agnostic variants) would be captured by an exome. Given the method limitations of exome sequencing, genomes offer better coverage (even for coding regions) than exomes do and are far superior for identifying CNVs and other structural variants [7]. All that said, genome data are costly to store and process computationally, and this should be considered alongside the benefits to having access of noncoding data, particularly if those data are mostly ignored.

Pane- based approaches that restrict analyses to clinically relevant genes clearly have merit, yet 26% of diagnoses made through the 100,000 Genomes Project were not on the original gene panel applied [8]. Therefore, complementary approaches that look beyond gene panels are warranted. However, this must be balanced with the potential of increasing the number of variants that require assessment by reporting laboratories. Currently, the GMS assess every variant that is in a ‘green’ gene in the PanelApp [4] gene panel applied, regardless of in silico predictions. Metrics such as CADD [9], REVEL [10], and SpliceAI [11] can help reduce noise, facilitating the assessment of variants across a wider spectrum of genes without too much additional burden. We sought to exploit this principle by adopting a panel agnostic approach that filters variants of **Hi**gh **P**athogenic **Po**tential (HiPPo) across the exome by utilising in silico prediction scores, allele frequency, and ClinVar [12] (Figure 2).

This study compares two different filtering approaches: one applied to exome sequencing data performed in a research setting, and another applied to genome sequencing performed on the same patients through the GMS in a clinical setting. To exome data, we apply a gene-agnostic approach, HiPPo, and compare the diagnostic yield of this approach with the strategy applied by the GMS. We aim to improve upon both the efficiency and diagnostic rates of current GMS standards whilst trying to minimise the number of variants requiring assessment by clinical laboratories.

## 2. Methods

### 2.1. Recruitment and Patient Demographics

Clinical geneticists at University Hospital Southampton were invited to recruit patients and families with suspected monogenic disease to a research study ‘*Use of NGS technologies for resolving clinical phenotypes*’ (IRAS: 212945; REC: 17/YH/0069). Recruited individuals were eligible for a research exome through the Center for Mendelian Genomics [13] at the Broad Institute.

Twenty-five individuals from eight families recruited and consented to the research exome study were also recruited for genome sequencing on the NHS through the GMS, facilitating a parallel comparison study (Figure 3) and providing an opportunity to evaluate these two sequencing and analysis strategies. All participants consented to their data being shared.

For the research exome study, patient phenotypes were extracted by a single researcher from the clinical notes and recorded as Human Phenotype Ontology (HPO) terms in a manually encrypted database. The patients’ clinicians also separately recorded HPO terms when requesting the GMS genome sequencing test. Both clinicians and researchers were blinded to each other’s curated HPO terms. The family structures of the 8 families (7 trios and a quad), individual IDs, and phenotypes are described in Table 1.

### 2.2. Research Exome Sequencing and Pipeline

Following quality control of the DNA from the 25 samples, the mother (FAM_4_12) in the family comprising a quad of parents and monozygotic twins (FAM_4) had insufficient DNA quality, and we were unable to obtain a repeat sample in time for inclusion in the research exome portion of this study. However, this participant had genome sequencing through the GMS. In the GMS, quads are sequenced as two separate trios. Therefore, family FAM_4 was exome sequenced without maternal data (father, twin A, twin B) for the research portion of the study but was genome-sequenced through the GMS as two separate trios (mother, father, twin A) and (mother, father, twin B).

A total of 24 samples from 8 families met the quality standards necessary for research exome sequencing at the Broad Institute (Appendix A). Libraries from DNA samples were created with an Illumina exome capture (37 Mb target) and sequenced on a NovaSeq 6000 machine using the NovaSeq XP workflow to cover >85% of targets at >20×, comparable to ~55× mean coverage. The samples underwent QC as previously described and were processed through the GATK best practices pipeline [14]. The samples were joint-called with >15,000 other samples and added to seqr v1.0 [15] (https://seqr.broadinstitute.org), an exome/genome analysis software hosted on the cloud platform Terra (https://app.terra.bio).

### 2.3. Genomic Medicine Service Pipeline

In total, 25 patients in 8 families were consented for GMS clinical genome sequencing; however, as one family (FAM_4) comprised a quad, the parents were sequenced with each child as two independent trios. Sequencing was performed on an Illumina NovaSeq 6000 machine, with ≥95% of the autosomal genome covered at ≥15× calculated from reads with mapping quality >10 and >85 × 10^9^ bases with Q ≥ 30, after removing duplicate reads and overlapping bases after adaptor and quality trimming. Cross-sample contamination was checked using VerifyBamID, and samples with >3% contamination failed QC. Sequencing alignment was performed using the DRAGEN aligner, with ALT-aware mapping and variant calling to improve specificity. Detection of small variants (single-nucleotide variants (SNVs) and indels) and CNVs were performed using the DRAGEN small variant caller and DRAGEN CNV, respectively. Short tandem repeat expansions were detected using ExpansionHunter (v2.5.6) as part of the DRAGEN software. The DRAGEN software v3.2.22 was used for alignment and variant calling, and structural variants were detected using Manta (v1.5).

### 2.4. Data Analysis

Different filtering strategies were applied to the research exome and the GMS genome data (Table 2). The research exome adopted the HiPPo strategy, and the GMS adopted a panel-based approach. The research filtering strategy collates and filters more information than the GMS approach. The GMS does not consider cohort allele frequency, ClinVar status or in silico metrics in its filtering strategy (Table 2).

### 2.5. Research Exome Analysis

For the research exome, each family was analysed as a unit to utilise segregation data. We applied the same de novo/dominant and recessive filtering strategies to all families, applying a gene-agnostic filtering strategy by selecting variants with the highest pathogenic potential (HiPPo) using allele frequency, in silico prediction scores, and ClinVar (Table 2). We later restricted the HiPPo strategy to GenCC [16] genes with a definitive or strong disease association.

### 2.6. Reporting on Exome Variants

Variants remaining following HiPPo filtering were reviewed in seqr [15] using a wealth of inbuilt annotations. Variants that did not meet any of the below exclusion criteria were considered ‘reportable’ and returned to the referring clinician following the application of ACMG-AMP guidelines [17]. Because the exome data were obtained in a research setting, we were able to include novel discoveries which would not meet diagnostic criteria in a clinical setting. Any novel discoveries were discussed with the referring clinician before submission to the Matchmaker Exchange (MME) [18,19,20].

Exclusion criteria:1.The variant was heterozygous in a known autosomal recessive disease gene, and no second hit (coding or noncoding) was identified;2.The variant was found in a disease gene and was not associated with the phenotype presented by the patient, as assessed using OMIM [21], GenCC [16] and the medical literature and the variant is not likely pathogenic/pathogenic in ClinVar [12];3.The variant is in a known disease gene, but that gene is poorly expressed, as indicated in GTEx [22], in the tissue relevant to the patient’s phenotype or in an exon of the gene with poor expression, as determined by the per base expression metric, pext [23];4.The variant was in a novel gene (currently unassociated with disease), and the gene is poorly expressed in the relevant disease tissue, as indicated in GTEx [22], or the gene is explicitly not involved in the relevant biological pathway, as evidenced in Monarch [24];5.A predicted loss-of-function (LoF) variant as called by Variant Effect Predictor [25] that was deemed as ‘not LoF’ or ‘likely not LoF’ after application of LoF manual curation guidance, as recommended by Karczewski et al. [26];6.The variant appeared artefactual upon visualisation of the read data in Integrative Genomics Viewer (IGV) [27].

### 2.7. Taking Novel Exome Candidates forward

Where the referring clinician agreed, candidate variants in novel genes were submitted by the researcher to MME, sharing anonymised genotype and phenotype data. Any potential matches were discussed in detail with the patient’s clinician, and explicit consent was obtained from the participants prior to joining the case series.

### 2.8. GMS Clinical Genome Analysis Pipeline

Variants called through the GMS pipeline were prefiltered on mode of inheritance, quality, and allele frequency. These variants were then restricted to ‘green’ genes on the preselected PanelApp [4] gene panels for review (Table 2). A complementary gene-agnostic filter was also applied to the data, which included all de novo variants and Exomiser [6] top 3 ranked coding variants (of any quality). Variants passing filtering were returned to the Wessex Regional Genetics Laboratory for reporting.

### 2.9. Reporting of GMS Genome Variants

GMS variant classification was carried out according to the ACMG/AMP guidelines with ACGS [28] modifications. This included an assessment of the gene-phenotype match based on the HPO [29] terms supplied. Variants in genes with no known disease association (determined using OMIM [21], HGMDPro, ClinGen [30] and PanelApp [4]) were discounted and not assessed. Classified variants were reported according to standard ACMG/AMP guidelines, i.e., pathogenic and likely pathogenic variants were always reported, and variants of uncertain significance (VUS) were only reported if there was significant evidence for pathogenicity and/or with the prior agreement of the clinician following a multidisciplinary team discussion (typically via email). 

### 2.10. Comparison of Two Filtering Approaches

We compared the diagnostic yield and the number of variants requiring assessment after variant filtering for both the research exome HiPPo approach (which included novel discoveries) and GMS clinical genome panel-based approach, which was restricted to reporting variants that met diagnostic standards only. Specifically, we counted the number of variants passing HiPPo filtering criteria in the research exome study and compared these with the number of Tier 1 and 2 variants for the same patients’ GMS genome results, in addition to the ‘gene agnostic’ variants (de novo and Exomiser [6] Top 3 hits) as provided in an anonymised spreadsheet by the Wessex Regional Genetics Laboratory. We omitted CNVs since these were not assessed in the exome data, and no diagnoses were made from structural variants in the GMS clinical genome data. We then compared the variants reported from the research exome with the variants interpreted and reported by the NHS on the patient’s GMS genome report. For the GMS, the reporting threshold is high with novel genes and nearly all VUS not reported. However as part of the GMS, patients have their deidentified data deposited into a genomics library for researchers to access. Therefore, to test the efficiency of the methods applied, we calculated the diagnostic rate per number of variants assessed across the cohort.

## 3. Results

### 3.1. GMS Clinical Genome Analysis Strategy

In the eight families who underwent GMS clinical genome sequencing, a total of 77 single-nucleotide and indel variants were returned for analysis as ‘Tiered variants’, including the gene-agnostic variants (Exomiser and de novo variants). A further 108 CNVs passed filtering. Five variants in total from four patients were included in the final reports issued by the NHS: two diagnoses, one variant of uncertain significance, and compound heterozygous variants (pathogenic and VUS); all five reported variants were also identified by HiPPo in the research exome (Table 3). 

### 3.2. Research Exome HiPPo Strategy

HiPPo identified a total of 109 variants (Appendix A) from eight families (eight trios) passing filtering criteria as outlined in Table 2. However, one family, FAM_4, comprising a mother, father and monozygotic twins, was sequenced as a trio (father, twin A and twin B) in the research exome study as there was insufficient maternal DNA. For the genome performed through the GMS, there was available maternal DNA, and thus, the twin daughters were sequenced as separate trios, with the parents sequenced twice in accordance with GMS policy. This meant more variants were identified in the research exome than the GMS genome (68 vs. 11, respectively) for this family, given that no maternal DNA was available for segregation analysis in the exome.

Of the 109 variants identified by HiPPo across the eight families, 38 variants were in genes reported as definitive or strong evidence for disease association, as classified by GenCC.

In addition to the two pathogenic variants identified by the GMS and deemed causal, HiPPo identified a further pathogenic variant in a known disease gene (*ABCC8*), representing a partial diagnosis that was filtered out by the GMS strategy due to not being on the chosen gene panel. HiPPo also identified compound heterozygous variants in a known disease gene, *INTS1*, in participant FAM_2_4, which is known to cause an autosomal recessive neurodevelopmental disorder with cataracts, poor growth, and dysmorphic facies (MIM: 618571). These variants were discounted by the GMS as weak VUS with limited evidence but remain under review by the clinical team. 

HiPPo detected a further six VUS in five novel (currently unassociated with disease) genes, in addition to the same compound heterozygous variants in *SDCCAG8* and the VUS in *HMGB1* reported by the GMS (Table 3). In total, the research exome identified 109 variants using HiPPo, of which 38/109 (34.9%) were in GenCC disease genes. After the application of exclusion criteria to all HiPPo variants, independent of GenCC disease status, a total of 14 variants from the research exome were curated against ACMG/AMP criteria and returned, as shown in Table 4.

On average, more HPO terms were recorded in the research exome study compared to the GMS genome (Table 1 and Figure 4) although this was not statistically significant (*p*-value = 0.1, Wilcox signed rank test).

When comparing the eight families who underwent parallel research exome and GMS clinical genome sequencing, we removed one family (FAM_4) from analysis as the mother was not sequenced in the research study but was sequenced by the GMS. There was no statistical difference between the number of variants (excluding CNVs) assessed by the GMS panel-based strategy and the HiPPo method (*p*-value = 0.35, Wilcoxon signed rank test), although HiPPo identified more reportable variants (Table 4), of which six variants in five unique genes were taken forward as candidates to MME. Three of these variants were identified but discounted by the GMS as disease gene discovery is outside of the remit of clinical reporting. However, when restricting the HiPPo analysis to GenCC strong and definitive genes, there was a statistical difference between groups (*p*-value = 0.022, Wilcoxon signed rank test), with the research study assessing fewer variants overall (Figure 4) yet still identifying an additional pathogenic variant in *ABCC8* that did not pass filtering thresholds by the GMS.

### 3.3. Comparison between Exome Study and GMS Clinical Genome Results

Table 3 (7.31%) compared with the panel-based GMS strategy (2/63 (3.17%)), although the result was not significant (*p*-value = 0.39, Fisher’s exact test). When limiting HiPPo analysis to GenCC disease genes, the diagnostic rate per variant assessed improved further to 3/15 (20%) (Figure 4); *p*-value = 0.06. The reportable variant rate per number of variants assessed was higher for the HiPPo approach when limited to GenCC disease genes (12/15 (80.0%)) compared with the panel-based GMS strategy (5/63 (7.93%)).

## 4. Discussion

Genome sequencing is available as a clinical test on the NHS through the GMS. Following sequencing, data are filtered by a preselected gene panel chosen by the referring clinician, in addition to CNVs overlapping the panel applied, de novo variants, and Exomiser [6] top three ranked variants. This predominantly ‘panel-based’ approach attempts to minimise noise and efficiently identify pathogenic variants in disease-relevant genes. 

However, panel-based strategies are not without limitations. PanelApp [4] is open source, but gene reviews and updates of the approved gene content rely on volunteer efforts and come with a significant lag time. Panels represent a snapshot in time, and their application is contingent on clinicians selecting the optimal gene panel(s) with variable levels of genetics training. This is particularly problematic for clinicians in nongenetics specialities lacking adequate familiarity with gene panel selection. If the “wrong” panel is chosen, pathogenic variants can easily be missed. With only 20% of rare disease patients receiving a diagnosis through the 100,000 Genomes Project [1] (the precursor to the UK’s GMS), there is a clear need to investigate variants beyond a limited gene list but without significantly increasing the number of variants for review.

This study compares the GMS’ data analysis filtering strategy using genome sequencing to a gene-agnostic HiPPo approach targeting variants with high pathogenic potential as applied to exome sequencing in a research setting. Twenty-four individuals from eight families underwent parallel clinical genome and research exome sequencing, providing an opportunity to compare different filtering approaches. With many factors influencing differences between the research and NHS studies, such as different technical pipelines, targeted capture, and timescales, the fairest comparison of the efficiency of the two approaches was the number of variants that required review following filtering and the corresponding diagnostic rates. On average, the research exome study reviewed fewer variants than the GMS, yet it identified more diagnostic variants, although this was not statistically significant (*p*-value = 0.35). The number of reportable variants per variant assessed was higher for HiPPo (29.3%) versus the GMS (7.9%). However, the threshold for what constituted a reportable variant differed between the research exome and the GMS genome strategies. The research exome reported on variants that would not be reportable in the current NHS setting, notably variants in novel disease genes and variants of uncertain significance, although it is worth noting that some international diagnostic labs do report variants in novel genes. However, when restricting the exome HiPPo filtering approach to GenCC disease genes (genes strongly associated with disease that would be reportable in the NHS setting), statistically fewer variants required assessment when compared with the GMS’ panel-based approach (Wilcoxon signed rank *p*-value = 0.022). Despite this, more pathogenic variants were identified, including a pathogenic variant in *ABCC8* representing a partial diagnosis, which was missed by the GMS as it was not on the selected gene panel. For the eight families undergoing parallel exome and genome sequencing, the GenCC disease gene HiPPo analysis strategy identified 15 variants that required further assessment, compared with 41 variants for the GMS approach. Overall, the diagnostic rate per number of variants assessed between the GenCC disease gene HiPPo analysis and the GMS’ panel-based approach was 3/15 (20%) vs. 2/63 (3%), respectively. Although our sample size is modest, there is a strong argument that genotype-to-phenotype methods focused on variants with high pathogenic potential in known disease genes could prove more effective and less resource-intensive than panel-based approaches despite covering a wider range of the genome. Indeed, in the GMS, very few Tier 2 variants are actually reported, meaning that Tier 1 + HiPPo may be an efficient alternative strategy and could also be used to prioritise the interpretation of gene-agnostic variants and/or determine which should be reported and/or taken to multidisciplinary team meetings. There is also a further argument that genome sequencing is not being optimally utilised by the NHS due to resource limitations and that exome sequencing may prove similarly effective; however, this comparison is beyond the scope of this limited study, whereby no pathogenic CNVs were identified, and a time-cost analysis could not be fairly undertaken. 

The number of HPO terms did not vary significantly between those selected for the research study versus those submitted by clinicians working in the NHS (*p*-value = 0.1) (Table 1). A recent study by Kingsmore et al. [31] showed that more HPO terms may not increase diagnostic yield but a more focused list of key terms may support analysis more effectively.

In total, HiPPo identified three diagnoses (compared with two diagnoses by the GMS) and a further 10 unique variants of interest in seven unique genes, of which three genes were discounted by the GMS (Table 4) as they did not meet the threshold for clinical reporting. In FAM_2_4, HiPPo identified compound heterozygous variants in *INTS1* (7:1480876G:C and 7:1497193:C:G), a disease gene associated with an autosomal recessive disorder (MIM: 618571) presenting with cataracts, poor growth, developmental delay, and dysmorphic facies. Whilst FAM_2_4 shares some features with the *INTS1*-related syndrome, he does not have cataracts and is large (with his weight tracking along the 99th percentile) as opposed to being small. These variants are being reviewed by his clinical team.

In FAM_8_23, both HiPPo and the GMS identified compound heterozygous variants (one pathogenic and one VUS) in *SDCCAG8* (1:243341070:TG:T and 1:243378799:A:G), a disease gene associated with an autosomal recessive retinal-renal ciliopathy (MIM: 615993 and MIM: 613615). These variants have been discussed at length with the clinical team and are not felt to explain the nephrotic phenotype. On renal biopsy, the patient had immature glomerular development diffuse foot process effacement on electron microscopy, which is inconsistent with a retinal-renal ciliopathy. Furthermore, there were additional inconsistent features, including microcephaly, cerebellopontine hypoplasia and functional asplenia. In the same individual, we identified a de novo variant in *ZNF91*. Through MME, we are collaborating with a group performing functional studies on this gene, whereby they also have a patient with microcephaly and nephrotic syndrome.

In total, we submitted six variants in five novel genes to MME from the exome study, which is beyond the remit of the NHS diagnostic capacity. We had no matches for *PKD1L3* and *FOXB2.* In addition to *ZNF91* (as described above in FAM_9_26), we matched with collaborators working on *HMGB1* (de novo variant found in FAM_1_1) and *ADGRB2* (de novo variant found in monozygotic twin sisters FAM_4_13 and FAM_4_10). In 2021, a paper was published on *HMGB1*-predicted loss-of-function variants in six patients [32]. Common features included developmental delay, language delay, microcephaly, obesity and dysmorphic features, some of which overlap with FAM_1_1. This variant has been returned to the patient’s clinician, and we have put them directly in touch with the authors of the 2021 paper [32] for an ongoing collaboration. Whilst the *HMGB1* variant was also reported as a VUS through the GMS, there is no time provision for clinicians to consider and follow-up on any unreported novel candidates. Furthermore, most de novo candidates in novel genes are disregarded by the GMS and so are seldom investigated further. That said, anonymised patient data are eventually deposited in the Genomics England Research Environment, meaning that novel variants may be identified and later investigated through research.

In 2017, a paper was published in Human Mutation describing a missense variant in *ADGRB2* in a patient presenting with developmental delay and progressive spastic paraparesis, features shared with identical twins FAM_4_13 and FAM_4_10 harbouring a de novo pLoF in the same gene [33]. The authors showed that their specific variant demonstrated a gain in function. We have contacted the authors of the paper and are now directly working with them to model our variant in vitro and in vivo.

### Limitations

This study is small, representing 25 individuals from eight families, of which 24 participants received parallel exome and genome sequencing, enabling us to compare two different filtering strategies. Inevitably, a larger study is needed to test the value of gene-agnostic approaches utilising pathogenicity scores compared with gene panel approaches. This is not currently possible within the NHS as we do not have routine access to data that would enable a direct comparison, such as the clinical panels selected for scrutiny, the tiered variants returned to the diagnostic laboratories, or the final clinical report. For the current study, this was only possible because we had local patient research consent to access these specific data from the GMS systems. We are mindful that we compared an exome with a genome, however, QC was excellent, and the panel-based strategy of the GMS essentially limits data analysis to exonic regions. We were further mindful that the two arms of the study used different technical pipelines, i.e., DRAGEN vs. GATK and tool versions may further contribute to differences in the performance of filtering strategies.

Data analysed in a research setting are not comparable with data analysed for diagnostic purposes as the threshold for variant follow-up and investigation differs in a clinical setting, with inconsistency in reporting on novel discoveries. It is important not to conflate a potential novel discovery identified in the research setting with a ‘missed’ diagnosis. That said, the research exome picked up a pathogenic variant missed by the GMS due to being outside the applied gene panels.

Our study was further biased by predominantly neurodevelopmental phenotypes due to the nature of intellectual disability (ID) being overrepresented in referrals for GMS genome sequencing. The biggest bias with ID is enrichment for de novo variants in this type of disorder, although both the GMS filtering approach and HiPPo assess de novo variants, so we would not expect the performance to change drastically.

No pathogenic variants were identified by GMS clinical genome sequencing that was not captured by the research exome, although a larger sample size is needed to test the diagnostic uplift gained from structural variants detected using genome sequencing versus potential missed diagnoses from using panel-based approaches. 

## 5. Conclusions

This study compared a gene-agnostic filtering strategy called HiPPo as applied to research exome data with a gene panel-based analysis strategy applied to genome sequencing data. Despite HiPPo being pan-exomic, a similar number of variants were assessed per patient to the panel-based strategy of the GMS and more variants of interest were identified; this includes a pathogenic variant in *ACDCC8* and de novo variants in three novel genes, whereby case series and functional experiments are underway, which highlights the added potential that research studies can offer. When restricting HiPPo to GenCC disease genes, statistically fewer variants required assessment to identify the same diagnoses as identified by the GMS (20% vs. 3%, respectively), representing a greater diagnostic yield per variant assessed. This preliminary work suggests that panel-based approaches are limited and that they could be improved by incorporating specific variant prioritisation metrics. Despite the limited sample size, we believe our study is an invaluable first step in demonstrating proof of concept that alternative analytical strategies have value for genome sequencing within the NHS.

## Figures and Tables

**Figure 1 healthcare-11-03179-f001:**
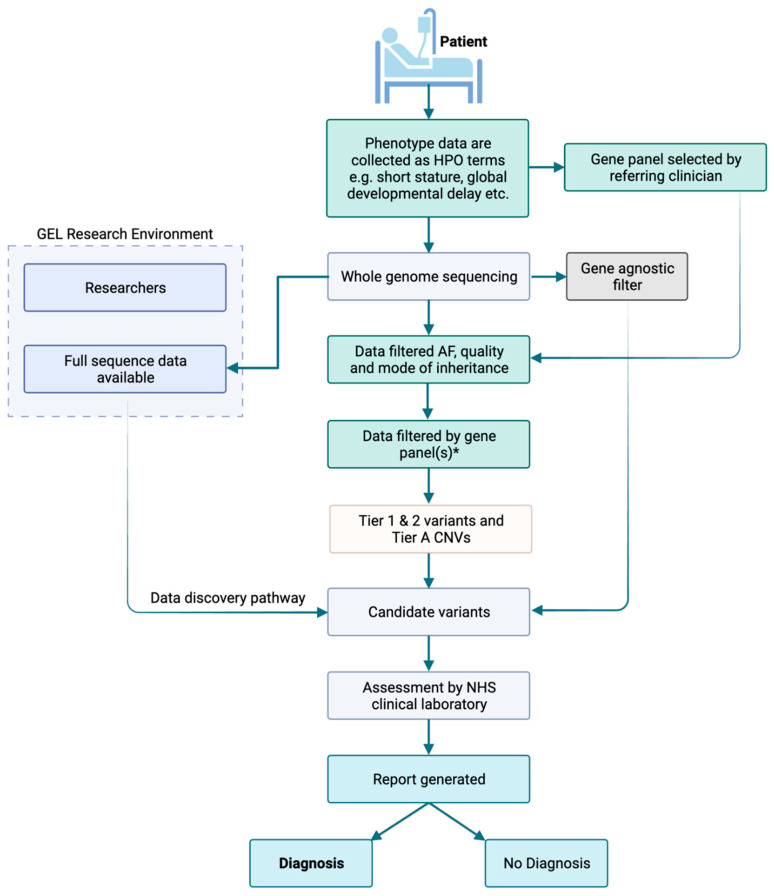
Genomic Medicine Service workflow for genome sequencing on the NHS. Tier 1 variants are defined as predicted loss-of-function variants or de novo variants in a green gene on the PanelApp gene panel(s) applied. Tier 2 variants are defined as coding variants +/− 8 bp (excluding synonymous) on any transcript in the panel applied. Synonymous variants affecting splicing are ignored. The gene-agnostic filter includes top three Exomiser rank variant with score of ≥0.95 and any de novo (coding) variant. Tier A is defined by a CNV (>10 KB) overlapping a ClinGen curated pathogenic region relevant to a panel applied or a CNV overlapping with a green gene in the panel applied. Anonymised sequencing data are available for some patients in the Genomics England (GEL) Research Environment. * Gene panels are selected using GEL PanelApp by the referring clinician.

**Figure 2 healthcare-11-03179-f002:**
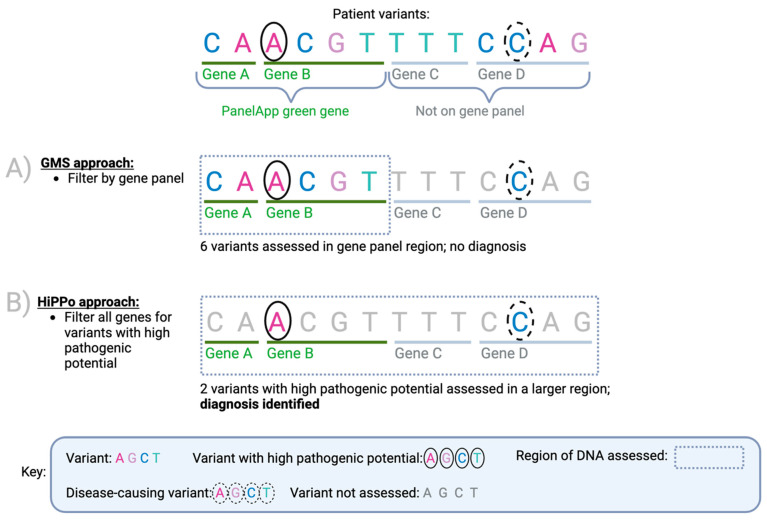
A proposed method for improving diagnostic yield and efficiency. Comparison of the current NHS approach versus our proposed method HiPPo for an example case (FAM 6). Variants are identified by comparing a patient’s DNA against a human genome reference. In this example, there is a pathogenic variant (dashed circle) within the identified list of variants. To minimise the number of variants assessed, the NHS has adopted a method (**A**) that looks at small regions of the DNA (a panel of genes) and assesses the variants within that region. If the causal variant is in a region of the DNA not assessed, then the diagnosis is missed. Our revised approach (**B**) captures a larger region of DNA (including all genes) but only looks at variants predicted to be damaging or submitted as P/LP to ClinVar (black circle). As a result, a larger area of DNA is assessed whilst assessing fewer variants overall. This aims to result in a higher diagnostic rate per number of variants assessed despite analysing a larger region of the genome than typically applied in a gene panel.

**Figure 3 healthcare-11-03179-f003:**
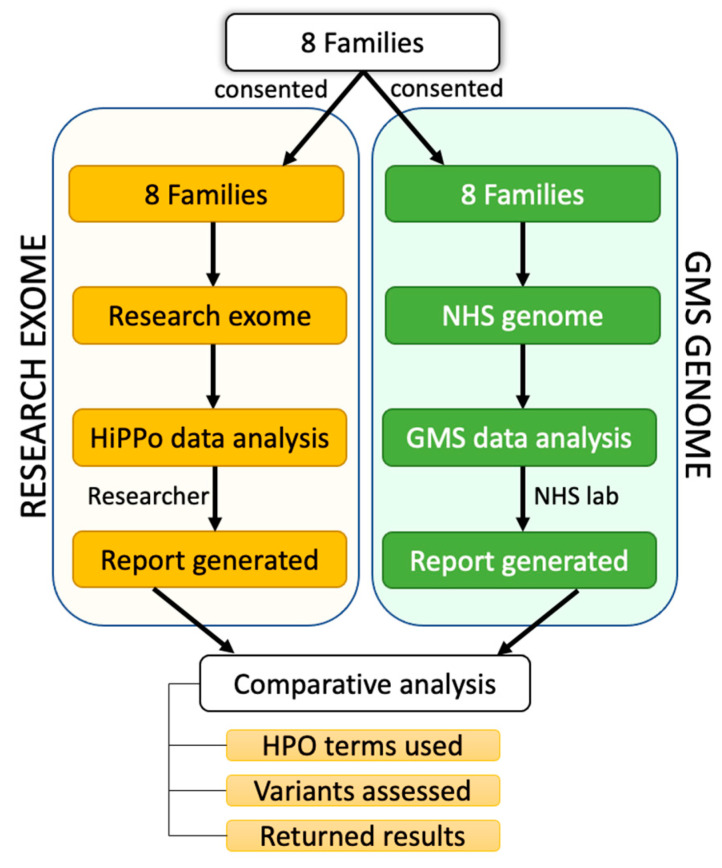
Overview of patient recruitment and analysis. Eight families were recruited for parallel GMS clinical genome sequencing and research exome sequencing. Different data analysis strategies were applied to the exome (HiPPo) vs. genome sequencing data (adopting a panel-based strategy as outlined by the GMS). Variants reported were compared between analysis strategies, including the Human Phenotype Ontology (HPO) terms used, number of variants assessed, and results reported.

**Figure 4 healthcare-11-03179-f004:**
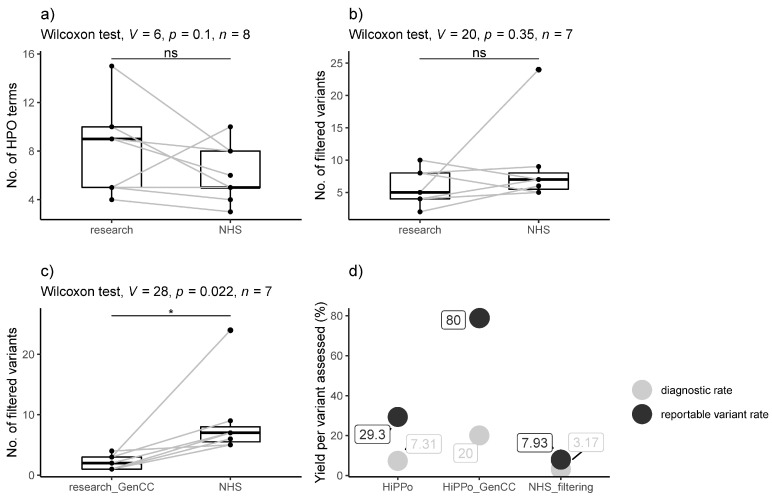
Comparison of results between the research exome and clinical genome (NHS) sequencing. (**a**) Number of HPO terms recorded between the exome and genome studies. (**b**) Number of variants assessed by the NHS reporting laboratory following GMS genome sequencing versus number of variants passing HiPPo filtering (in any gene) in the exome study. (**c**) Number of variants assessed by the NHS reporting laboratory following GMS genome sequencing versus the number of filtered HiPPo variants in GenCC disease genes assessed by the exome study. (**d**) Plot showing the diagnostic rate per variant assessed and the reported variant rate per variant assessed for the HiPPo research approach, HiPPo restricted to GenCC disease genes approach, and the GMS panel-based filtering strategy. * *p* < 0.05.

**Table 1 healthcare-11-03179-t001:** Samples and phenotypes of patients recruited for a parallel research exome and NHS genome.

Samples					Clinical Data				
Fam_ID	Pro_ID	Pat_ID	Mat_ID	Sib_ID	Age	Sex	WES Phenotype—HPO Terms Identified from Clinical Notes	WGS Phenotype—HPO Terms Identified by Clinician	WGS—Gene Panel Applied
FAM_1	1	2	3		0–5	M	Gastroesophageal reflux, **Myopia**, Delayed eruption of primary teeth, Triangular face, Prominent forehead, **Cow milk allergy**, **Egg allergy**, **Nut food product allergy**, **Sacral dimple**, **Clinodactyly of the 5th finger**, **Short 5th toe**, **2–3 toe syndactyly**, Mild global developmental delay, Delayed speech and language development, Oligohydramnios	Global developmental delay, Delayed speech and language development, Triangular face, Prominent forehead, Feeding difficulties, Delayed gross motor development, Oligohydramnios, Delayed eruption of primary teeth.	Intellectual disability (**R.29.4**), Congenital malformation and dysmorphism syndromes (**R27.3**), Skeletal dysplasia (**R104.3**), Likely inborn error of metabolism (**R98.2**)
FAM_1	2				41–45	M	unaffected	unaffected	
FAM_1	3				41–45	F	unaffected	unaffected	
FAM_2	4	5	6		6–10	M	**Simple ear**, **Astigmatism**, Obesity, **Patchy hypo- and hyperpigmentation**, **2–3 toe syndactyly**, **Short finger**, **Specific learning disability**, Global developmental delay, Intellectual disability, **Delayed speech and language development**	Chronic otitis media, Obesity, Severe intellectual disability, Autistic behaviour, Global developmental delay	Intellectual disability (**R.29.4**), severe early-onset obesity (**R149.1**)
FAM_2	5				31–35	M	unaffected	unaffected	
FAM_2	6				31–35	F	unaffected	unaffected	
FAM_3	7	8	9		6–10	F	**Low-set ears**, **Hypermetropia**, Abnormality of the palmar creases, **Broad distal phalanges of all fingers**, **Shallow orbits**, **Cranial asymmetry**, Plagiocephaly, **Mild global developmental delay**, Intellectual disability	Thin upper lip vermillion, **Long philtrum**, **Downslanted palpebral fissures**, Deep palmar crease, Intellectual disability, Plagiocephaly	Intellectual disability (**R29.4**)
FAM_3	8				61–65	M	unaffected	unaffected	
FAM_3	9				46–50	F	unaffected	unaffected	
FAM_4	10	11	12	13	46–50	F	**Delayed ability to walk**, **Delayed speech and language development**, Spastic paraparesis, Global developmental delay	Global developmental delay, **Intellectual disability**, and Spastic paraparesis	Intellectual disability (**R29.4**)
FAM_4	11				76–80	M	unaffected	unaffected	
FAM_4	12				76–80	F	unaffected	unaffected	
FAM_4	13				46–50	F	**Delayed ability to walk**, **Delayed speech and language development**, Seizure, Spastic paraparesis, **Global developmental delay**	Developmental delay, **Intellectual disability**, Spastic paraparesis, and Seizure	Intellectual disability (**R29.4**)
FAM_5	14	15	16		0–5	F	Prominent forehead, Low hanging columella, **Prominent fingertip pads**, **Preauricular pit**, **Hypopigmented macule**, Frontal bossing, **Flat occiput**, **Joint hypermobility**, **Confluent hyperintensity of cerebral white matter on MRI**, Mild global developmental delay, **Polydipsia**	Prominent forehead, Moderate global developmental delay, **Relative macrocephaly**, **Anxiety**, Low hanging columella	Intellectual disability (**R29.4**)
FAM_5	15				26–30	M	unaffected	unaffected	
FAM_5	16				21–25	F	unaffected	unaffected	
FAM_6	17	18	19		0–5	F	Epicanthic folds, **Joint hypermobility**, Global developmental delay, **Intellectual disability**, Increased nuchal translucency	Global developmental delay, Increased prenatal nuchal translucency, **Short toenails**, Epicanthic folds	Intellectual disability (**R29.4**)
FAM_6	18				31–35	M	unaffected	unaffected	
FAM_6	19				31–35	F	unaffected	unaffected	
FAM_7	20	21	22		6–10	M	**Hypertelorism**, Bilateral polymicrogyria, Global developmental delay, Delayed speech and language development, **Delayed fine motor development**, **Delayed gross motor development**, Focal seizures, Generalised seizures, Intellectual disability	Focal seizures, Generalised seizures, **Infantile encephalopathy**, Polymicrogyria, Delayed speech and language development, Severe intellectual disability, Global developmental delay	Early-onset or syndromic epilepsy (**R59.3**), Cerebral malformation (**R87.3**)
FAM_7	21				36–40	M	unaffected	unaffected	
FAM_7	22				36–40	F	unaffected	unaffected	
FAM_8	23	24	25		0–5 *	F	Microphthalmia, Cataract, Retinal dystrophy, Congenital nephrotic syndrome, Microcephaly	**Intrauterine growth restriction**, Microcephaly, Congenital nephrotic syndrome, **Renal failure**, Bilateral congenital cataract, **Cerebellopontine hypoplasia**, Retinal dysfunction, **Thrombocytopaenia**, **Giant platelets**, **Howell–Jolly bodies**	Congenital malformation and dysmorphic syndromes (**R27**), Structural eye disease (**R36**), Unexplained paediatric onset end-stage renal disease (**R257**), Cerebellar anomalies (**R84**), Severe microcephaly (**R88**), Proteinuric renal disease (**R195**)
FAM_8	24				36–40	F	unaffected	unaffected	
FAM_8	25				41–45	M	unaffected	unaffected	

Discrepancies between phenotypes underlined in bold. Ages given in age ranges. Fam_ID = Family ID, Mat_ID = Maternal ID, Pat_ID = Paternal ID, Pro_ID = Proband ID, Sib_ID = Sibling ID. All IDs are fully anonymised for publication. * Patient deceased.

**Table 2 healthcare-11-03179-t002:** Filtering criteria for the research exome and NHS genome.

	Research Exome HiPPo Strategy		NHS Genome Panel-Based Strategy	
	Dominant	Recessive	Dominant	Recessive
**Inheritance**	De novo/dominant search	Recessive search ^&^	De novo/dominant search	Recessive search ^&^
**AF (gnomAD exomes, gnomAD genomes, TOPMED *, ExAC, 1000 g)**	<0.001	<0.05	<0.001	<0.01
**Cohort^ AF**	<0.01	<0.01	No filter applied	No filter applied
**Variant type**	All coding +/− 20 bp, excluding synonymous, on any transcript	All coding +/− 20 bp, excluding synonymous, on any transcript	All coding +/− 8 bp on any transcript, excluding synonymous	All coding +/− 8 bp on any transcript, excluding synonymous
**SpliceAI (for splicing variants)**	>0.2	>0.2	No filter applied	No filter applied
**CADD (all variants)**	>15	>15	No filter applied	No filter applied
**ClinVar**	Remove benign/likely benign	Remove benign/likely benign	No filter applied	No filter applied
**Genes**	All genes and later restricted to GenCC definitive and strong genes	All genes and later restricted to GenCC definitive and strong genes	Green in PanelApp Panel(s)	Green in PanelApp Panel(s)
**Allele balance**	>0.2	>0.2	N/A	N/A
**Genotype Quality**	>40	>40	>30	>30
**QC**	all variants	all variants	pass	pass
**Other**	Pathogenic variants in ClinVar retained even if in unaffected parents	N/A	In any gene: Exomiser top 3 rank variant (coding) with score of ≥0.95 or any de novo (coding)	In any gene: Exomiser top 3 rank variant (coding) with score of ≥0.95 or any de novo (coding)
**SV/CNV**	Not assessed	Not assessed	CNV (>10 KB) overlaps a ClinGen curated pathogenic region relevant to a panel applied, or the CNV overlaps with a green gene in the panel applied.	CNV (>10 KB) overlaps a ClinGen curated pathogenic region relevant to a panel applied, or the CNV overlaps with a green gene in the panel applied.

Comparison of filtering criteria between the research exome and NHS genome. AF—maximum allele frequency across any population, QC—quality control, N/A—not applicable. * TOPMED allele frequency was only applied to the research exome. ^ Cohort AF is the frequency of any given variant as a frequency of the total number of individuals in that cohort (>15,000 individuals for the research study). ^&^ Recessive search includes X-linked recessive disorders.

**Table 3 healthcare-11-03179-t003:** Comparison of variants reported by the research exome sequencing study vs. the GMS genome sequencing.

Samples		Research Exome				GMS Genome					
FamID	ProID	Reported Variants	Status	No. HiPPo Variants	No. HiPPo Variants in GenCC Genes	Reported Variants	No. of Variants Passing Filtering	De Novo	Exomiser	Additional HiPPo Variants	GMS Interpretation of HiPPo Variants
FAM_1	1	**VUS:** *HMGB1*: 13:30462666:CT:C; c.342del; p.Gly115GlufsTer37 (frameshift, de novo).	Potential new disease gene, submitted to MME. Variant also detected by NHS.	8	3	**VUS:** *HMGB1*: 13:30462666:CT:C; c.342del; p.Gly115GlufsTer37	9	*REST * *HMGB1*	*1. HMGB1* *2. KDM4* *3. ROBO1*	None	N/A
FAM_2	4	**VUS:** *INTS1*: 7:1480876:G:C; c.3908C > G; p.Thr1303Ser (missense).	Phenotype partially fitting with disease gene—undergoing clinical review.	4	4	No variants reported	5	*None*	*1. KDM5A* *2. RPS3A* *3. COL16A1*	*INTS1*—VUS × 2	*INTS1* (Tier 2) discounted as weak evidence
		**VUS:** *INTS1*: 7:1497193:C:G; c.1547G > C; p.Cys516Ser (missense).									
		(Variants in trans)									
FAM_3	7	**Pathogenic:** *PPP1CB*: 2:28776944:C:G; c.146C > G; p.Pro49Arg (missense, de novo).	Confirmed diagnosis (also detected by NHS).	4	2	**Pathogenic:** *PPP1CB*: 2:28776944:C:G; c.146C > G; p.Pro49Arg	7	*MYO7B* *PPP1CB * *EXOC7*	*1. PPP1CB* *2. SELENBP1* *3. EFCAB11*	None	N/A
FAM_4	10	**VUS:** *ADGRB2*: 1:31731030:G:A; c.4150C > T; p.Arg1384Ter (de novo, nonsense).	Potential new disease gene. Confirmed de novo by Sanger sequencing and in identical twin (FAM_4_13). Functional work underway.	68	23	No variants reported	14	*ADGRB2* *CRNN * *PCDHB7* *NFYB * *PIEZO1*	*1. FBXO46* *2. CEP290* *3. NFYB*	*ADGRB2*—VUS × 2	De novo (*ADGRB2*) variant discounted as in novel gene
FAM_4	13	**VUS:** *ADGRB2:* 1:31731030:G:A; c.4150C > T; p.Arg1384Ter (de novo, stop gained).	The same variant is present in identical twin (FAM_4_10)			No variants reported					
FAM_5	14	**Pathogenic:** *ABCC8*: 11:17413408:G:A; c.2464C > T; p.Gln822Ter (nonsense, inherited from parent)	Clinically agreed as partial diagnosis.	8	1	No variants reported	5	*GOLGA8T*	*1. PTPRF* *2. NPHP4* *3. PRKDC*	*ABCC8*—**Pathogenic**	*ABCC8* not analysed as untiered and gene absent from R29 panel
FAM_6	17	**Pathogenic:** *CHAMP1*: 13:114325034:C:T; c.1192C > T; p.Arg398Ter (de novo, nonsense).	Confirmed diagnosis (also detected by NHS).	2	1	**Pathogenic:** *CHAMP1*: 13:114325034:C:T; c.1192C > T; p.Arg398Ter	6	*KRTAP5-5*	*1. CHAMP1* *2. MDK* *3. CRAC2RA*	None	N/A
FAM_7	20	**VUS:** *FOXB2*: 9:77020700:A:G; c.1046A > G; p.Lys349Arg (missense, de novo). **VUS:** PKD1L3: 16:71951734:T:G; c.3020A > C; p.Glu1007Ala (missense).	Both FOXB2 and PFK1L3 are potential novel disease genes and have been submitted to MME.	10	1	No variants reported	7	*FOXB2* *RP1L1*	*1. IGFN1* *2. ZXDA* *3. CADNA1F*	*FOXB2*—VUS *PKD1L3*—VUS × 2	*FOXB2* de novo variant—Discounted *PKD1L3*—Not analysed—Tier 3; Exomiser rank 33
		**VUS:** *PKD1L3*: 16:71951734:T:G; c.3020A > C; p.Glu1007Ala (missense). *PFK1L3* variants are in trans.									
FAM_8	23	**VUS:** *ZNF91*: 19:23361341:G:C; c.1638C > G; p.Tyr546Ter (de novo, nonsense).	*ZNF91* is a novel disease gene. A group is working on this gene, and we have joined their case series. The *SDCCAG8* variants are in trans but are not felt to fit with the clinical phenotype.	5	3	**VUS:** *SDCCAG8*: 14:92449109:A > C, c.1552A > G, p.Arg518Gly (missense). **Pathogenic:** *SDCCAG8*: 1:243341070:TG>T, c.1255del, p.Glu419ArgfsTer43 (frameshift).	24	*ZNF91* *ZNF91*	*1. RIN3* *2. ERAP2* *3. ZNF91*	None	*ZNF91* variant discounted as no established disease association
		**VUS:** *SDCCAG8*: 14:92449109:A > C, c.1552A > G, p.Arg518Gly (missense).									
		**Pathogenic:** *SDCCAG8*: 1:243341070:TG > T, c.1255del, p.Glu419ArgfsTer43 (frameshift).									

FamID—Family ID; MME—matchmaker exchange; N/A—not applicable; ProID—Proband ID; VUS—variant of uncertain significance.

**Table 4 healthcare-11-03179-t004:** Details of 14 variants reported by the research exome study meeting prioritisation criteria.

Variant	Gene	Consequence	Gnomad	Cadd	Revel	Hgvsc	Hgvsp	ClinVar	ACMG	FamID	ProbandID	P_AC	Sample_2	S2_AC	Sample_3	S3_AC	Returned by GMS?
13:30462666:CT:C	*HMGB1*	frameshift	0	38		ENST00000341423.9:c.342del	p.Gly115GlufsTer37		VUS	FAM_1	1	1	2	0	3	0	Yes
7:1480876:G:C	*INTS1*	missense	5.56^−4^	23.5	0.243	ENST00000404767.7:c.3908C > G	p.Thr1303Ser		VUS	FAM_2	4	1	5	1	6	0	No
7:1497193:C:G	*INTS1*	missense	7.76^−5^	24	0.315	ENST00000404767.7:c.1547G > C	p.Cys516Ser		VUS	FAM_2	4	1	5	0	6	1	No
**2:28776944:C:G**	** *PPP1CB* **	**missense**	**0**	**26.7**	**0.438**	**ENST00000395366.2:c.146C > G**	**p.Pro49Arg**	**P**	**P**	**FAM_3**	**7**	**1**	**8**	**0**	**9**	**0**	**Yes**
1:31731030:G:A	*ADGRB2*	stop_gained	0	38		ENST00000373655.6:c.4150C > T	p.Arg1384Ter		VUS	FAM_4	13	1	11	0	13	1	No
1:31731030:G:A	*ADGRB2*	stop_gained	0	38		ENST00000373655.6:c.4150C > T	p.Arg1384Ter		VUS	FAM_4	10	1	11	0	13	1	No
**13:114325034:C:T**	** *CHAMP1* **	**stop_gained**	**0**	**35**		**ENST00000643483.1:c.1192C > T**	**p.Arg398Ter**	**P**	**P**	**FAM_6**	**17**	**1**	**18**	**0**	**19**	**0**	**Yes**
**11:17413408:G:A**	** *ABCC8* **	**stop_gained**	**0**	**43**		**ENST00000302539.9:c.2464C > T**	**p.Gln822Ter**		**LP**	**FAM_5**	**14**	**1**	**15**	**0**	**19**	**1**	**No**
16:71951734:T:G	*PKD1L3*	missense	5.09^−4^	23.1		ENST00000620267.1:c.3020A > C	p.Glu1007Ala		VUS	FAM_7	20	1	21	1	22	0	No
16:71973386:C:T	*PKD1L3*	missense	1.02^−4^	22		ENST00000620267.1:c.1891G > A	p.Ala631Thr		VUS	FAM_7	20	1	21	0	22	1	No
9:77020700:A:G	FOXB2	missense	0	25.3	0.534	ENST00000376708.1:c.1046A > G	p.Lys349Arg		VUS	FAM_7	20	1	21	0	22	0	No
19:23361341:G:C	*ZNF91*	stop_gained	0	32		ENST00000300619.11:c.1638C > G	p.Tyr546Ter		VUS	FAM_8	23	1	24	0	25	0	No
1:243341070:TG:T	*SDCCAG8*	frameshift	0	26		ENST00000366541.7:c.1255del	p.Glu419ArgfsTer43		P	FAM_8	23	1	24	1	25	0	Yes
1:243378799:A:G	*SDCCAG8*	missense	9.55^−5^	22.2	0.195	ENST00000366541.7:c.1552A > G	p.Arg518Gly	VUS	VUS	FAM_8	23	1	24	0	25	1	Yes

Families separated by colour. FamID—Family ID; hgvsc—HGVS coding consequence; hgvsp—HGVS protein consequence; LP—likely pathogenic; P—pathogenic; P_AC—proband allele count; S2_AC—sample2 allele count; S3_AC—sample3 allele count; VUS—variant of uncertain significance. Sample_2 and sample_3 refer to parental DNA.

## Data Availability

Data are contained within the article and Appendix A.

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
