# Peer review of "A Panel-Agnostic Strategy ‘HiPPo’ Improves Diagnostic Efficiency in the UK Genomic Medicine Service"

_healthcare, 2023, doi:10.3390/healthcare11243179_

Round 1

Reviewer 1 Report

Comments and Suggestions for Authors

The accurate identification of pathogenic variants including novel ones is very important for genomics used for clinical analysis. The authors of this study performed a pilot study of 24 participants from eight families to compare their exome-based approach, HiPPo, to the panel and whole genome-based approach to identify the pathogenic variants. The authors have found more variants that require clinical assessment with their approach which is cost-effective and it does require whole genome sequencing data.

Following are a few suggestions and comments for this study.

Major comments

1. This study is based on a very small number of samples (authors also acknowledged in the limitation section) and the difference in identified variant count is not that significant. However, both methods used different tools for variant discovery (GATK and DRAGEN). So it may not be a fair comparison as false positive/false negative cases may differ in both tools. Also, the use of the latest versions of the tools may also make a bigger difference due to ongoing improvements of variant calling algorithms of both methods. 

2. The authors should have added the variant calling method/pipelines in the Method section. The use of different parameters and filtering techniques may give different results. So it would be great to have all the details so that the claims can be explained properly.

3. The filtering methods used in both approaches (Table 2) are different e.g. "No filter applied" for most of the cases for the panel-based strategy. The authors could elaborate on that portion and explain why same filtering methods are not applied. 

4. The copy number variation (CNV) analysis in this study may need more explanations. The first section of the result mentions that the 108 CNVs passed filtering, so a comparison of CNV discovery among the two approaches and their utility could be added. Also, the methods used to identify CNVs should be added to the method section. 

Minor comments

1. The 2nd line of the abstract states "...variants in novel gene..". Should it be "novel variants in genes.."?

2. Many citation numbers are after the full stop e.g. "..including in the NHS.(1) However..". They should be before the end of the sentence. 

3. Different formatting of texts in Fig 1 legend.

4. There are no yellow highlighted circles in Figure 2, but it states "...is a pathogenic variant (yellow highlighted circle) within..."

5. The heading "Research exome sequencing and pipeline" and some others should be formatted according to other similar headings

6. Use "Supplementary Table 1" instead of "Supplementary Data Table 1"

7. The first few columns of Table 1 should be formatted properly.

8. Citation for the "...2021 paper for ongoing.." is required.

Comments on the Quality of English Language

Minor edits are required

Author Response

The accurate identification of pathogenic variants including novel ones is very important for genomics used for clinical analysis. The authors of this study performed a pilot study of 24 participants from eight families to compare their exome-based approach, HiPPo, to the panel and whole genome-based approach to identify the pathogenic variants. The authors have found more variants that require clinical assessment with their approach which is cost-effective and it does require whole genome sequencing data.

We thank the reviewer for their considered comments on our manuscript. We are certain their suggestions, which we address below, will improve the quality of our manuscript.

Following are a few suggestions and comments for this study.

Major comments

  1. This study is based on a very small number of samples (authors also acknowledged in the limitation section) and the difference in identified variant count is not that significant. However, both methods used different tools for variant discovery (GATK and DRAGEN). So it may not be a fair comparison as false positive/false negative cases may differ in both tools. Also, the use of the latest versions of the tools may also make a bigger difference due to ongoing improvements of variant calling algorithms of both methods. 

The reviewer makes a fair point. Regretfully, the use of the variant discovery tools was out of our hands and as the reviewer mentions, the number of variants did not drastically vary between studies. We do mention in the discussion section that “With many factors influencing differences between the research and NHS studies, such as different technical pipelines, targeted capture, and timescales, the fairest comparison of efficiency of the two approaches was the number of variants that required review following filtering and the corresponding diagnostic rates.” However, we have expanded the limitation section to highlight the valid point being made.

  1. The authors should have added the variant calling method/pipelines in the Method section. The use of different parameters and filtering techniques may give different results. So it would be great to have all the details so that the claims can be explained properly.

The variant calling method pipeline for both the research and NHS arms are already included in the Methods section. The methods are included under the heading “Research exome and sequencing pipeline” and also “Genomic Medicine Service pipeline”. The details are also below for reference:

“25 patients in 8 families were consented for GMS clinical genome sequencing; however, as one family (FAM_4) comprised a quad, the parents were sequenced with each child as two independent trios. Sequencing was performed on an Illumina NovaSeq 6000 machine, with ≥95% of the autosomal genome covered at ≥15x calculated from reads with mapping quality >10 and >85x10^9 bases with Q≥30, after removing duplicate reads and overlapping bases after adaptor and quality trimming. Cross-sample contamination was checked using VerifyBamID and samples with >3% contamination failed QC. Sequencing alignment was performed using the DRAGEN aligner, with ALT-aware mapping and variant calling to improve specificity. Detection of small variants (single nucleotide variants (SNVs) and indels) and CNVs were performed using the DRAGEN small variant caller and DRAGEN CNV respectively. Short tandem repeat expansions were detected using ExpansionHunter (v2.5.6) as part of the DRAGEN software. The DRAGEN software v3.2.22 was used for alignment and variant calling and structural variants were detected using Manta (v1.5).”

A total of 24 samples from 8 families met quality standards necessary for research exome sequencing at the Broad Institute (Supplementary Table 1). Libraries from DNA samples were created with an Illumina exome capture (37 Mb target) and sequenced on a NovaSeq 6000 machine using the NovaSeq XP workflow to cover >85% of targets at >20x, comparable to ~55x mean coverage. The samples underwent QC as previously described and were processed through the GATK best practices pipeline.(14) The samples were joint called with >15,000 other samples and added to seqr(15) (https://seqr.broadinstitute.org), an exome/genome analysis software hosted on the cloud platform Terra (https://app.terra.bio). “

  1. The filtering methods used in both approaches (Table 2) are different e.g. "No filter applied" for most of the cases for the panel-based strategy. The authors could elaborate on that portion and explain why same filtering methods are not applied. 

Thank you to the reviewer for highlighting that this needed further clarification. We have added a sentence to say thatThe research filtering strategy collates and filters on more information than the GMS approach. The GMS does not consider cohort allele frequency, ClinVar status, nor in silico metrics in its filtering strategy (Table 2).”

  1. The copy number variation (CNV) analysis in this study may need more explanations. The first section of the result mentions that the 108 CNVs passed filtering, so a comparison of CNV discovery among the two approaches and their utility could be added. Also, the methods used to identify CNVs should be added to the method section. 

The methods used to identify CNVs is already included in the methods section. Please see the section called “Genomic Medicine Service pipeline” for details. The specifics are also below:

“Detection of small variants (single nucleotide variants (SNVs) and indels) and CNVs were performed using the DRAGEN small variant caller and DRAGEN CNV respectively. Short tandem repeat expansions were detected using ExpansionHunter (v2.5.6) as part of the DRAGEN software. The DRAGEN software v3.2.22 was used for alignment and variant calling and structural variants were detected using Manta (v1.5).”

We were conscious that CNVs were not included in the exome side of the study, however comparing CNVs from an exome to a genome would introduce even further bias in what is already a limited study. We were conscious to make the comment that There is also a further argument that genome sequencing is not being optimally utilised by the NHS due to resource limitations and that exome sequencing may prove similarly effective; however, this comparison is beyond the scope of this limited study, whereby no pathogenic CNVs were identified, and a time-cost-analysis could not be fairly undertaken.

In the limitations section we further say that: No pathogenic variants were identified by GMS clinical genome sequencing that were not captured by the research exome, although a larger sample size is needed to test the diagnostic uplift gained from structural variants detected using genome sequencing versus potential missed diagnoses from using panel-based approaches.”

Minor comments

  1. The 2nd line of the abstract states "...variants in novel gene..". Should it be "novel variants in genes.."?

We have clarified this to say: “Variants in genes without established disease-gene relationships”.

  1. Many citation numbers are after the full stop e.g. "..including in the NHS.(1) However..". They should be before the end of the sentence.

We would like to leave this to be an editorial decision; some journals prefer before and some after.

  1. Different formatting of texts in Fig 1 legend.

Apologies, this appears to be a conversion issue. We have gone through the manuscript and fixed formatting errors.

  1. There are no yellow highlighted circles in Figure 2, but it states "...is a pathogenic variant (yellow highlighted circle) within..."

This has been amended to “dashed circle”.

  1. The heading "Research exome sequencing and pipeline" and some others should be formatted according to other similar headings

We have amended the headings to be italicised.

  1. Use "Supplementary Table 1" instead of "Supplementary Data Table 1"

Thank you, we have changed this.

  1. The first few columns of Table 1 should be formatted properly.

Many apologies; there has were some formatting issues introduced after submission. We have amended Table 1 but will leave it up to the editorial team to finalise how they would like the table presented.

  1. Citation for the "...2021 paper for ongoing.." is required.

This refers to the 2021 paper written in the sentence above. We have re-added the citation again for clarity.

Reviewer 2 Report

Comments and Suggestions for Authors

Authors developed a High Pathogenic Potential (HiPPo) strategy using exome data from 24 participants in 8 families to improve diagnostic rate per sequence variant assessed. Several de novo variants were identified and associated with disease phenotypes. Although such associations have not yet been confirmed with cause-effect relationship, this new strategy facilitates disease sequence variant discovery compared with whole genome sequencing. The methodology and findings are sound, and limitations of this new strategy are clearly indicated. Only editorial revision below is needed.

Delete the first paragraph of the Results section “25 individuals from 8 families were consented for a GMS clinical genome on the NHS…. Therefore, there were a total of 24 individuals in 8 263 families who completed parallel research exome and GMS clinical genome sequencing”, which is redundant to several paragraphs in preceding Methods section. 

Author Response

Authors developed a High Pathogenic Potential (HiPPo) strategy using exome data from 24 participants in 8 families to improve diagnostic rate per sequence variant assessed. Several de novo variants were identified and associated with disease phenotypes. Although such associations have not yet been confirmed with cause-effect relationship, this new strategy facilitates disease sequence variant discovery compared with whole genome sequencing. The methodology and findings are sound, and limitations of this new strategy are clearly indicated. Only editorial revision below is needed.

We thank the reviewer for their time reading and reviewing our manuscript. We are delighted that the manuscript was reviewed so positive.

Delete the first paragraph of the Results section “25 individuals from 8 families were consented for a GMS clinical genome on the NHS…. Therefore, there were a total of 24 individuals in 8 263 families who completed parallel research exome and GMS clinical genome sequencing”, which is redundant to several paragraphs in preceding Methods section. 

Thank you for the suggestion. We have deleted the first paragraph of the results section as suggested.

Reviewer 3 Report

Comments and Suggestions for Authors

The manuscript written by Seaby et al. presents a relevant method for exome analysis, based on pathogenicity data. The method is well described and the results are adequately discussed. I have minor questions/suggestions for the authors:

1. Have they tried to execute the method in X-linked disorders?

2. Is it possible to include non-familiar cases in the analysis (i.e., proband without family history)?

3. Is any of the families from an underrepresented ethnicity in public genome databases such as gnomAD? How does the method behave considering different minor allelic frequencies?

4. Minor point: in Table 3, gene ZNF91 must be italicized.

Author Response

The manuscript written by Seaby et al. presents a relevant method for exome analysis, based on pathogenicity data. The method is well described and the results are adequately discussed. I have minor questions/suggestions for the authors:

We thank the reviewer for taking time to review our manuscript.

  1. Have they tried to execute the method in X-linked disorders?

Our recessive search included X-linked recessive disorders. We have modified Table 2 to reflect this and highlighted in the legend.

  1. Is it possible to include non-familiar cases in the analysis (i.e., proband without family history)?

We think the reviewer is referring ‘proband-only’ cases, whereby parental data is unknown. This would be an interesting idea, however criteria for inclusion in the Genomic Medicine Service is for analysis of a trio specifically. This means that our comparator group will always be a trio rather than any proband-only cases at this stage.

  1. Is any of the families from an underrepresented ethnicity in public genome databases such as gnomAD? How does the method behave considering different minor allelic frequencies?

Yes, one family is, however the journal limits the sharing of ethnicity specific data. We apply the maximum allele frequency across any population, which overcomes any issues with minor allele frequencies varying between ethnic populations. We have amended the legend of Table 2 to reflect this.

  1. Minor point: in Table 3, gene ZNF91 must be italicized.

Thank you, we have amended.